# Effects of Local Vasodilators and the Autonomic Nervous System on Microcirculation and Mitochondrial Function in Septic Rats

**DOI:** 10.3390/ijms25179305

**Published:** 2024-08-28

**Authors:** Carsten Marcus, Claudia Hansen, Charlotte Schlimgen, Jeanne Eitner-Pchalek, Jan Schulz, Stefan Hof, Anne Kuebart, Richard Truse, Christian Vollmer, Inge Bauer, Olaf Picker, Anna Herminghaus

**Affiliations:** Department of Anesthesiology, University Hospital Duesseldorf, Moorenstrasse 5, 40225 Duesseldorf, Germany; carsten.marcus@med.uni-duesseldorf.de (C.M.); claudia.hansen1991@gmail.com (C.H.); charlotte.schlimgen@hhu.de (C.S.); jeanne.eitner-pchalek@uk-koeln.de (J.E.-P.); janandschulz@web.de (J.S.); stefan.hof@med.uni-duesseldorf.de (S.H.); anne.kuebart@med.uni-duesseldorf.de (A.K.); richard.truse@med.uni-duesseldorf.de (R.T.); christian.vollmer@med.uni-duesseldorf.de (C.V.); inge.bauer@med.uni-duesseldorf.de (I.B.); picker@med.uni-duesseldorf.de (O.P.)

**Keywords:** sepsis, animal model, CASP, microcirculation, mitochondria, vasodilators, nitroglycerin, iloprost, carbachol

## Abstract

Systemic vasodilating agents like nitroglycerin (NG) or iloprost (Ilo) show beneficial effects on intestinal microcirculation during sepsis, which could be attenuated by activation of the sympathetic nervous system or systemic side effects of vasodilating agents. This exploratory study aimed to investigate the effects of topically administered vasodilators and the parasympathetic drug carbachol on colonic microcirculatory oxygenation (µHbO_2_), blood flow (µFlow) and mitochondrial respiration. A total of 120 male Wistar rats were randomly assigned to twelve groups and underwent either colon ascendens stent peritonitis (CASP) or sham surgery. After 24 h, animals received the following therapeutic regimes: (1) balanced full electrolyte solution, (2) carbachol, (3) NG, (4) Ilo, (5) NG + carbachol, and (6) Ilo + carbachol. Mitochondrial respiration was measured in colon homogenates by respirometry. In sham animals, NG (−13.1%*) and Ilo (−10.5%*) led to a decrease in µHbO_2_. Additional application of carbachol abolished this effect (NG + carbachol: −4.0%, non-significant; Ilo + carbachol: −1.4%, non-significant). In sepsis, carbachol reduced µHbO_2_ when applied alone (−10.5%*) or in combination with NG (−17.6%*). Thus, the direction and degree of this effect depend on the initial pathophysiologic condition.

## 1. Introduction

The fact that sepsis and septic shock is one of the most challenging tasks in intensive care medicine is underpinned by the high incidence of 18% and mortality rate of up to 42% [1,2]. The driving factors of sepsis are, among others, the malfunctioning gastrointestinal microcirculation and mitochondrial dysfunction. Both can lead to a failure in the mucosal barrier, resulting in the induction of a sterile inflammation by the local gut-associated lymph tissue or a direct entry of endotoxins in the systemic circulation, which massively engraves sepsis and consecutively causes multi-organ dysfunction syndrome. Despite intensive research, the main therapeutic strategy of sepsis mainly consists of eradication of the focus and supportive therapy, including antibiotics, fluid resuscitation, and application of vasoactive substances [3]. Nevertheless, mortality remains high. Thus, new therapeutic approaches are necessary to improve the efficacy of the treatment and to reduce the high lethality.

Studies have shown the positive effects of direct vasodilative agents such as nitric oxide (NO) donors like nitroglycerin (NG) and prostacyclin agonists, e.g., iloprost (Ilo), on the mucosal oxygenation of the colon. These effects are presumably mediated not only by their vasodilatory but also by cytoprotective effects on the intestinal barrier [4,5,6,7]. In addition, NG and Ilo have shown a positive effect on the healing of anastomosis after surgery, which can be of special interest in abdominal sepsis [8,9]. Another positive effect of both drugs is their impact on mitochondrial function causing a reduction of the maximal mitochondrial respiration and improvement of the efficacy of oxidative phosphorylation in colon homogenates in vitro [10].

Unfortunately, systemic application of vasodilating drugs can lead to negative effects like cardiovascular depression, especially under septic conditions. In this context topical application of these drugs might avoid systemic adverse side effects and enable to profit from potential beneficial effects at the target organ.

Of note, topical therapy is already widely established. Inhalative Ilo is used in pulmonary arterial hypertension [11]. Truse et al. showed that the topical application of Ilo and NG has a beneficial effect on gastric oxygenation during hemorrhagic shock without compromising systemic hemodynamic variables [12].

Because sepsis is accompanied by an activation of the sympathetic nervous system, causing increased systemic vascular resistance, especially by increased precapillary vasoconstriction, the positive effect of NG and Ilo on the microcirculation could be abolished [13]. Moreover, the dysregulation of the microcirculation occurs even before macrocirculatory abnormalities can be detected [14]. The direct-acting cholinergic drug carbachol activates the muscarinergic receptors, promoting the liberation of acetylcholine and, consecutively, vasodilatation. This may improve the microvascular circulation. An additional positive effect of carbachol might be a protective effect on the intestinal barrier, as shown in previous investigations [15,16].

In addition to microcirculation, mitochondrial function plays a key role in sepsis and aggravation thereof. During sepsis, various vasomotoric, hormonal and metabolic effects lead to mitochondrial dysfunction, which results in insufficient adenosine triphosphate (ATP) generation and consecutively aggravation of the sepsis in a multiple-organ dysfunction syndrome (MODS) [17]. Although NG and Ilo were shown to improve the efficacy of oxidative phosphorylation, in vitro studies investigating the effect on mitochondrial function in the colon of septic animals are lacking [10].

Taken together, the aim of this exploratory study was to investigate the influence of topical application of direct vasodilative agents and the effect of a modification of the autonomic balance by a local acting parasympathomimetic on the gastrointestinal oxygenation as well as colonic mitochondrial function under septic conditions. Positive effects of the topical therapy in the gastrointestinal tract might represent a therapeutic option, e.g., as an enteral nutrition augmentation or as oral administration in the treatment of septic patients in the ICU.

## 2. Results

### 2.1. Macrocirculation

There was a significant decrease in mean arterial pressure (MAP) after 60 min compared with baseline in each group, independent of the type of surgery (CASP or sham), except for the colon ascendes stent peritonitis (CASP) carbachol group (Table 1). Likewise, heart rate (HR) decreased after 60 min, except for the sham Ilo and CASP Ilo + carbachol group (Table 1). There were no significant differences in MAP or HR between the groups.

### 2.2. Microcirculation

#### 2.2.1. Effect of NG and Carbachol on Microcirculation in Sham Animals

In the non-septic control group, no significant change in microcirculatory oxygenation could be measured. Animals treated with NG had a significant decrease in µHbO_2_ compared with baseline values (Figure 1a sham nitroglycerin −13.1 ± 18.3%, *p* < 0.05 vs. baseline) after 60 min. This effect was abolished with additional carbachol application. The sole therapy of carbachol did not affect mucosal oxygenation. The perfusion (µFlow) did not change significantly during the experiment in the groups, also there was no statistically significant effect between the groups.

#### 2.2.2. Effect of Ilo and Carbachol on Microcirculation in Sham Animals

Similar to the NG groups, there was a decrease in microcirculatory oxygenation after 60 min in the Ilo group (Figure 2a, sham iloprost, −10.5 ± 9.7%, *p* < 0.05 vs. baseline). In combination with carbachol (Figure 2a, sham iloprost + carbachol), the effect was abolished. There was no change in the µFlow in either group and no change between the groups.

#### 2.2.3. Effect of NG and Carbachol on Microcirculation in Septic Animals

A significant change in microcirculatory oxygenation could not be seen in the control sepsis group. Beyond that, a local application of NG did not affect the µHbO_2_ of the colon as well. µHbO_2_ in septic animals treated with NG and carbachol showed a significant decrease compared with baseline after one hour (Figure 3a, CASP nitroglycerin carbachol −9.2 ± 24.5%, *p* < 0.05 vs. baseline). Similarly, a local monotherapy with carbachol caused a decline in µHbO_2_ after 60 min as well (Figure 3a CASP carbachol −10.5 ± 15.7%, *p* < 0.05 vs. baseline). The µFlow did not change significantly in any group compared with the baseline. There were no differences in µFlow and µHbO_2_ between the groups either (Figure 3b).

#### 2.2.4. Effect of Ilo and Carbachol on Microcirculation in Septic Animals

Under septic conditions, treatment with Ilo showed no effect on the microcirculatory oxygenation. The combination of Ilo and carbachol also had no effect on mucosal oxygenation. As already mentioned, the sole application of carbachol caused a decrease in µHbO_2_ after 60 min (Figure 4a CASP carbachol −10.5 ± 15.7%, *p* < 0.05 vs. baseline). Neither Ilo nor carbachol showed any effect on the µFlow or in comparison with the other groups.

### 2.3. Mitochondrial Respiration

#### 2.3.1. Effect of NG and Carbachol on Mitochondrial Respiration under Septic and Non-Septic Conditions

The treatment with NG, carbachol or a combination of both drugs did not affect the colonic mitochondrial function. Respiratory control index (RCI) and oxidative phosphorylation efficacy (ADP/O) stayed unchanged in all groups under septic and non-septic conditions (Figure 5a–h).

#### 2.3.2. Effect of Ilo and Carbachol on Mitochondrial Respiration under Septic and Non-Septic Conditions

The treatment with Ilo increased the RCI without affecting ADP/O for both complexes (I and II) after the sham operation (Figure 6a–d). Carbachol and the combination of both drugs showed no effect on the mitochondrial function under non-septic conditions (Figure 6a–d). Under septic conditions, none of the applied substances affected the mitochondrial function (Figure 6e–h).

The study was performed to evaluate the effect of topical therapy of NG, Ilo and carbachol on the colonic microcirculation and mitochondrial function in rats under septic and non-septic conditions. Our main results are as follows:Topical therapy of NG and Ilo, as well as additional carbachol, has no influence on the macrocirculatory variables heart rate and blood pressure compared with the control group.NG and Ilo cause a negative effect on the microcirculatory oxygenation after sterile laparotomy, which is not measurable in combination with carbachol.Under septic conditions, carbachol reduced µHbO_2_. This effect is abolished by additional therapy with Ilo.NG shows no effect on the colonic mitochondrial function neither under septic or non-septic conditions.Under non-septic conditions, Ilo improves the coupling between the respiratory chain and oxidative phosphorylation without affecting the efficacy of the latter. This effect is abolished during sepsis and through a combination with carbachol.

## 3. Discussion

The chosen colon ascendens stent peritonitis model is a well-established model of abdominal sepsis. This model induces moderate sepsis with slowly aggravating systemic inflammation by constant fecal leakage through the implanted stents. Several authors describe the CASP model as an animal model that closely mimics abdominal peritonitis in humans, and thus, this model is advantageous over other experimental sepsis models like cecal ligature and puncture (CLP) or lipopolysaccharide (LPS) injection [18,19] for the here described research question. In combination with visible peritonitis after the induction of surgery and reproducibly high IL6, IL-10 and TNF-alpha levels in previous studies with the same model, an abdominal infection of the animals is reliably detected [20,21,22].

The microcirculatory variables were measured at a penetration depth of 0.7 mm. This enabled an assessment of alterations in oxygenation and flow in the entire colonic wall (0.35 mm) [23]. As we depict a cross-section of the entity of capillary vessels smaller than 100 µm, the majority of the measured targets are postcapillary venules, as about 85% of blood circulation is pooled there. This comes in useful for our studies as this is the compartment that is mainly affected by septic changes in vascular resistance [24].

In this study, alterations in cardiovascular parameters were observed, specifically noting a significant decline in heart rate (HR) and mean arterial pressure (MAP) across most groups after 60 min. This decline, possibly induced by anesthesia, aligns with previous studies demonstrating anesthesia’s impact on these parameters [25]. The combined action of hypnotic and analgesic agents potentially induces bradycardia and hypotension, attributed to sympatholysis, but no substantial variance in this effect was detected between groups at the chosen time points (T0, T30, T60), reinforcing this hypothesis. Furthermore, the hypotension did most certainly not affect the microcirculation, as there were no significant correlations between hypotension and a decrease in microcirculatory oxygenation or perfusion in our studies.

The local application of NG, Ilo and carbachol aimed to mitigate the systemic effects of these drugs on circulation, with macrocirculatory parameters displaying no significant differences between the interventions and the control group, confirming our objective. Therefore, local therapy seems to be safe and free from systemic hemodynamic side effects. However, we only studied the effect of the substances on a small part of the gastrointestinal tract. Nevertheless, topical therapy in areas at risk of anastomoses is clinically feasible.

In this study, a negative influence of the vasodilators NG and Ilo on microcirculatory parameters after a sterile laparotomy is shown. Modulation of the autonomic balance with carbachol abolished the decrease in µHbO_2_. The decrease in oxygenation could be explained by either an increase in cellular oxygen utilization or a decrease in perfusion [26].

At first glance, a decrease in perfusion seems unlikely, as therapy with NG and Ilo should increase perfusion through vasodilation. In addition, decreased perfusion would present itself through a lower µFlow. Nevertheless, as the therapy with Ilo and NG primarily manifests itself in the arteries and veins, it is conceivable that it increases the precapillary oxygen loss there through vasodilatation in the sense of arteriovenous shunting [27,28]. However, it is questionable whether this effect is sufficient to ensure a significant reduction in µHbO_2_.

Cabrales et al. made use of an animal model to investigate pre- and post-capillary microvascular oxygen consumption in endothelial nitric oxide synthase (eNOS)-deficient mice [29]. Their findings suggest that the oxygen consumption in mice with inhibited NO synthesis is increased. However, they were also able to show that the increase in oxygen consumption is tissue-specific, as it only takes place in the tissue and not in the microcirculation. Interestingly, our experiments showed no impairment in oxygen utilization, as there were no significant changes in the mitochondrial measurements in either of these groups. A possible limitation of our measurements could be that the mitochondrial data are measured in colon homogenates in vitro, which could mask a difference in specific tissues in vivo.

In contrast, there was no effect of local application of vasodilators on the µHbO_2_ in septic animals. In sepsis, various endogenous mechanisms are involved in improving the imbalance between oxygen supply and demand. In this context, two vasoactive mechanisms leading to vasodilatation are in the foreground: the enhanced activation of inducible nitric oxide synthase (iNOS) and the cyclic adenosine monophosphate (cAMP) pathway. Therefore, we aimed to address these two pathways using NG and Ilo. It is conceivable that these pathways were already exhausted, explaining the missing effect after additional therapy with vasodilators. These results go along with a double-blinded, placebo-controlled trial in intensive care patients by the group of Boerma et al. in 2010, where no improvement of microcirculation after NG therapy could be found [30]. Another explanation of the absent effect of NG could be a nitrate tolerance or nitrate tachyphylaxis, a widely described decrease of the effect of NG in patients after a certain period of time, usually starting at 24 h after first admission. However, keeping in mind that the total duration of NG therapy in this study was one hour, this explanation seems unlikely. The data on nitrate tolerance in rats are contradictory, but the vast majority of studies showed it after a longer time period [31].

Contrary to the sham animals, modulation of the autonomic balance with carbachol led to a decrease in µHbO_2_ under septic conditions. Thus, the effect of vasodilators NG and Ilo and the effectivity of the autonomic modulation with carbachol seems to be dependent on initial conditions like sepsis or sterile laparotomy. Endothelial dysfunction in a septic environment can cause an uncoupling of the endothelial nitric oxide synthase (eNOS) of its physiological mechanism, meaning that eNOS produces superoxide anions (O_2_^−^) instead of NO. In the course of this uncoupling, oxidative stress causes the eNOS cofactor tetrahydrobiopterin (BH_4_) to oxidate to dihydrobiopterin (BH_2_). This different cofactor enhances the production of superoxides by a competitive antagonism with BH_4_ [32]. The now-produced superoxide anion causes a vicious cycle because the increased presence of O_2_^−^ results in increased production of more radical oxygen species (ROS). Additionally, it reacts with NO, leading to an increased amount of peroxynitrite (ONOO^−^), another ROS, increasing the amount of oxidative stress even further [33,34] (Figure 7).

Based on that, it seems plausible that our therapy with NG, which is basically a topical liberation of NO, has no vasoactive effect, as the growing amount of O_2_^−^ inactivates its vasodilating effect by reacting to peroxynitrite. In this context, a paradox effect of acetylcholine is described, meaning a vasoconstrictor instead of a dilator effect in an impaired endothelium [35]. This would explain the decrease in µHbO_2_ in the topical treatment with carbachol, as this cholinomimetic drug activates acetylcholine receptors. However, it must be mentioned that µFlow did not undergo any significant changes in our experiments, but the results showed a clear tendency towards a decrease.

Looking at the described molecular mechanisms, it is conceivable that the administration of Ilo with its G protein-coupled pathway can lead to vasodilatation through the liberation of cyclic adenosine monophosphate (cAMP). This could compensate for the vasoconstriction induced by carbachol and preserve the microcirculatory oxygenation, resulting in an unchanged µHbO_2_, as seen in this trial. Additionally, prostacyclin analogs like Ilo show an anti-oxidant effect, which may be beneficial under septic conditions regarding the uncoupling of eNOS [36].

Several researchers investigated the influence of NG and Ilo on intestinal circulation. Our results are opposite to those of Truse et al., who showed an improvement of impaired gastric microcirculation under topical therapy with NG and Ilo in hemorrhagic shock in dogs [12]. It is possible that the divergent results are caused by different pathophysiological states. As elaborated above, sepsis can cause endothelial dysfunction, which most likely will not develop into a short-term hemorrhage.

In this investigation, NG demonstrated no discernible impact on colonic mitochondrial function, regardless of septic or non-septic conditions. These findings contradict results obtained by Herminghaus et al., which revealed a positive influence of nitroglycerin on oxidative phosphorylation efficacy (ADP/O) in colon homogenates [10]. However, in contrast to this study, the study by Herminghaus et al. was conducted in tissue homogenates from healthy rats, using defined drug concentrations (25 µg/mL and 250 µg/mL NG) directly added to tissue homogenates. In our study, NG was topically applied to the colon lumen without assessing drug concentration within the colonic wall. It is plausible that due to dilution in the colon lumen and absorption through the colonic wall, the ultimate drug concentration in the target cells was lower than in the aforementioned in vitro study. Additionally, the difference between the in vitro experiments by Herminghaus et al. and our ex vivo effects should also not be underestimated, as well as the experimental setting, as the time of mitochondrial measurement in the in vitro study could be performed immediately after the experiment, as the substances were incubated with the mitochondria.

Contrarily, Ilo improved the coupling between the respiratory chain and oxidative phosphorylation, but solely under non-septic conditions. The coupling between the respiratory chain and the oxidative phosphorylation was improved because state 2 was lowered (the change was not significant). State 3 for complex I was lowered (not significant either), and complex II was similar to the control. The respiratory control index (RCI) is a calculated value, and the non-significant changes in state 2 and state 3 led to a significant change in RCI. This partly aligns with the findings by Herminghaus et al., demonstrating reduced mitochondrial function in maximal respiration (state 3) and oxidative phosphorylation efficacy (ADP/O) [10]. While they did not analyze the RCI, the reduced state 3 suggests a potential improvement in the RCI calculated from state 3 and state 2. Discrepancies between the results of these studies probably stem from differing designs, test environments and likely divergent drug concentrations, as detailed earlier. Intriguingly, the mitochondrial function was influenced by Ilo depending on the initial conditions. However, it exhibited no impact on colonic mitochondrial function during sepsis. A plausible explanation might be that mitochondria possess compensatory mechanisms toward maintaining cellular homeostasis to shield cells during adverse conditions like sepsis [37]. Herminghaus et al. illustrated improved hepatic mitochondrial function under mild and early sepsis compared with a sterile laparotomy, suggesting that further improvement might not occur if the mitochondrial function is already heightened by septic conditions. Notably, alterations in colonic mitochondria were not observed under septic conditions in that study [20]. However, the sepsis model utilized in our study differed slightly from the model used by Herminghaus et al. (two stents in colon ascendens vs. one stent).

### Future Work Directions and Study Limitations

Consequently, as the mechanisms governing Ilo’s effects on colonic mitochondrial function are variable and depending on the initial condition (septic vs. non-septic), further research for full comprehension of involved pathways is necessary. It could be conceivable that measurements of BH_4_ levels and enzymatic eNOS activity compared to sham surgery, as described by Crabtree et al., could strengthen our theory of eNOS-uncoupling [38]. A study investigating both values in abdominal sepsis could close a gap in the pathophysiology of this condition.

The doses used in our experiments were based on data from the literature, which in the case of carbachol, showed a positive effect on microcirculation in trauma in rats [39]. The fact that the dosages of our local therapy with NG and Ilo are taken from a trial in dogs is to be discussed as a limitation; however, no systemic but microcirculatory effect was found after application of the substances, which suggests that the substances show an effect to the extent intended in the trial design [12].

As we used a preclinical model in rats, the translation in human species could be influenced by species-specific variability. Additionally, the experimental setting cannot fully represent the multifactorial and dynamic processes in human sepsis to the full extent. Nevertheless, the CASP surgery is the closest model to simulate failure of surgical anastomosis and not just bacteriaemia like, for example, the LPS model.

Therefore, our study addressed a very common pathology in intensive care medicine, and our results show the effects of frequently used medication on gastrointestinal circulation and mitochondria.

## 4. Materials and Methods

### 4.1. Surgical Induction of Sepsis

All parts of this study were performed in accordance with NIH guidelines for animal care and reported in accordance with the ARRIVE guidelines. Experiments started after approval from the local Animal Care and Use Committee (Landesamt fuer Natur, Umwelt und Verbraucherschutz, Recklinghausen, Germany, Az. 84-02.04.2015.A538).

The entire experiment was completed in 120 male Wistar rats (320–380 g), which were provided by the breeding facility at the Central Animal Research Facility of the Heinrich-Heine-University Duesseldorf. CASP or sham surgery were performed to create sepsis or serve as a control group, respectively. The animals were randomly assigned to 12 groups (n = 10) receiving topical therapy of NG and Ilo, either with or without the addition of carbachol (Figure 8). In the CASP group, 8 animals died during 24 h of sepsis induction. Those animals were replaced and the experiments were repeated in a blinded manner.

The experiments were carried out in the laboratory of the Dept. of Anesthesiology, University Hospital Duesseldorf, Germany. CASP surgery was performed as described earlier [22]. Briefly, a median laparotomy of 2 cm was performed under general anesthesia (buprenorphine 0.1 mg·kg^−1^, sevoflurane 3.0–3.2% endexspiratory volume, FiO_2_ 0.5) and the cecum was located. The antimesenteric wall of the colon was penetrated at least 1 cm distal of the ileocecal valve with two 14 G stents (Vasofix^®^Safety, 14 G x 2″, B. Braun Melsungen AG, Melsungen, Germany), cut down to a length of 10 mm. The stents were placed about 6–8 mm deep and secured with a 6.0 Prolene suture (Ethicon Inc., Bridgewater, NJ, USA), granting a constant leak of feces in the abdominal cave. The peritoneal membrane was closed with a running suture and the skin with single simple interrupted sutures. The sham surgery consisted of opening the abdomen in the same way, but in contrast to CASP, a single stent was sutured to the antimesenterial wall of the cecum without perforating it.

After CASP and sham surgeries, the animals were kept in individual cages (Makrolon cage Typ III, surface area 825 cm^2^, height 15 cm, Ehret GmbH, Emmendingen, Germany), receiving analgesia (buprenorphine 0.05 mg·kg^−1^, TEMGESIC^®^, Eumedica Pharmaceuticals GmbH, Lörrach, Germany) every 8 h. Access to water and food ad libitum was provided as well as a 12 h day/night rhythm and a constant temperature (24 ± 2 °C) and humidity (50 ± 5%). Every 6 h, the animals underwent an earlier described protocol (Septic Rat Severity Score: SRSS-System) to determine the severity of sepsis and to monitor the animals with respect to their welfare (loss of body weight, appearance, spontaneous behavior, provoked behavior, breathing frequency, expiratory breathing sound, abdominal palpation and condition of droppings) [40]. The scoring of all animals was performed by the same investigator; if the score was 10 or more, the animal was euthanized with pentobarbital (600 mg·kg^−1^ i.p., sodium pentobarbital, Fagron GmbH & Co KG, Glinge Germany).

### 4.2. Assessment of Microcirculation

At 24 h after CASP or sham operations, the animals were anesthetized with pentobarbital (60 mg·kg^−1^ i.p.) and buprenorphine (0.1 mg·kg^−1^ s.c.) injection and placed on a heating pad. After a tracheotomy was performed, volume-controlled, pressure-limited mechanical ventilation (70 min^−1^, VT 1.8–2.5 mL, PAW < 17 cmH_2_O, FiO_2_ 0.3, FiN_2_ 0.7) was initiated to maintain normocapnia (pCO_2_ 45 ± 5 mmHg) (Inspira Advanced safety Ventilator, Harvard Apparatus GmbH, March-Hugstetten, Germany). Two 24 G catheters (Introcan Safety^®^-Winged, 24 G x 3/4″, B. Braun Melsungen Ag, Melsungen, Germany) were placed in the arteria carotis communis and the external jugular vein each. At baseline and following every 30 min, arterial blood gas analyses were performed to ensure normoventilation. Invasive blood pressure measurement was performed through the same arterial catheter, which also administered a continuous infusion of 4 mL balanced crystalloid solution Jonosteril (Jonosteril^®^ FRESENIUS KABI Deutschland GmbH, Bad Homburg, Germany) per hour to replace fluid loss. The maintenance of general anesthesia was performed by a continuous pentobarbital infusion (10 mg·kg^−1^) via the jugular vein catheter. A total of 2 mg of the muscle relaxant pancuronium (Pancuronium bromid, Inresa Arzneimittel GmbH, Freiburg, Germany) was injected at the beginning of the baseline period.

After catheterization, the animals were relaparotomized and a flexible light guide probe (O2C LW 2222, Lea Medizintechnik GmbH, Giessen, Germany; penetration depth 0.7 mm, tissue catchment area 7 mm^2^, separation 1 mm) was placed on the colon ascendens 1 cm next to the stent to measure intestinal microcirculation as described previously [21,41]. Briefly, using tissue-reflectance spectrophotometry, white light (450–1000 nm) is transmitted to the colonic wall, and the reflected light is analyzed. The fact that oxygenated hemoglobin absorbs wavelengths of about 900 nm and deoxygenated hemoglobin absorbs wavelengths of about 680 nm allows us to measure the saturation of the regional hemoglobin (µHbO_2_). As light is absorbed by vessels larger than 100 µm, only microcirculation is measured [42]. The perfusion is measured via laser Doppler flowmetry. In this process, the velocity of erythrocytes is calculated by changes in the Doppler frequency of the emitted light. The intensity of the emitted light is used to calculate the relative amount of erythrocytes [rHb], which is set in relation to the light without a change in Doppler frequency. This change in the number of erythrocytes is multiplied by the velocity, resulting in the flow of the capillaries (µFlow), which represents the perfusion. As the main part, with about 85% of the blood volume located in the venous system, mainly the postcapillary part of the circulation is analyzed, which is the critical partial pressure for hypoxia [43]. The quality of the signal is continuously online-evaluated throughout the experiments allowing a constant verification of the correct position of the probe tip. Starting at baseline, the µHbO_2_ and µFlow values are reported as the means of the last 5 min of the 30 min periods.

To apply the topical therapy, after 30 min of baseline recording, a stent was placed into the transition of cecum to colon ascendens. During 1 h NG ([0.025 µg/g bolus + 25 µg/g/h (25 µg/mL) infusion] nitrolingual infusion, G. Pohl-Boskamp Co.KG, Hohenlockstedt, Germany) or Ilo ([0.1 ng/g bolus + 0.02 ng/g/h (100 ng/mL) infusion], 5-cis iloprost, Cayman Chemicals, Ann Arbor, MI, USA), rats were infused either with or without additional carbachol ([0.06 μg/g bolus + 1.2 µg/g/h (120 μg/mL) infusion] Carbamylchlorine chloride C4382, Sigma-Aldrich, St. Louis, MO, USA). The dosages in our experiment were previously described in the literature as local therapy in animals [12,39]. Before application, drugs were diluted in Jonosteril. The investigator was blinded to the treatments. At the end of the experiments, the animals were euthanized by exsanguination via the arterial catheter.

### 4.3. Preparation of Colon Homogenates and Protein Level Measurements

Colon homogenates were prepared as described previously [10,20,44]. Briefly, freshly harvested colon tissue was placed in 4 °C cold isolation buffer (200 mM mannitol, 50 mM sucrose, 5 mM KH_2_PO_4_, 5 mM morpholinepropanesulfonic acid (MOPS), 1 mM EDTA, pH 7.15). They were then longitudinally opened and dried softly with a cotton compress to remove feces and muscus. After treatment with trypsin for 5 min on ice, tissue was placed in 4 °C isolation buffer containing 20 mg/mL BSA and protease inhibitors (cOmplete™ Protease Inhibitor Cocktail, Roche Life Science, Mannheim, Germany), minced into 2–3 mm^3^ pieces and homogenized (Potter-Elvehjem, 5 strokes, 2000 rpm).

Protein concentration in the tissue homogenates was measured using the Lowry method with bovine serum albumin as a standard [45].

### 4.4. Measurement of Mitochondrial Respiratory Rates

Measurement of the mitochondrial oxygen consumption was performed at 30 °C using a Clark-type electrode (model 782, Strathkelvin instruments, Glasgow, Scotland) as described before [20,40,46]. Tissue homogenates were suspended in respiration medium (130 mM KCl, 5 mM K_2_HPO_4_, 20 mM MOPS, 2.5 mM EGTA, 1 µM Na_4_P_2_O_7_, 2% BSA, pH 7.15) to yield a protein concentration of 6 mg/mL.

Mitochondrial state 2 respiration was recorded in the presence of either complex I substrates glutamate and malate (both 2.5 mM, G-M) or complex II substrate succinate (5 mM, S). The maximal mitochondrial respiration in state 3 was measured after the addition of ADP (50 µM). The respiratory control index (RCI) was calculated (state 3/state 2) to define the coupling between the electron transport system and oxidative phosphorylation. To reflect the efficacy of oxidative phosphorylation, the ADP/O ratio was calculated from the amount of ADP added and O_2_ consumption. The average oxygen consumption was calculated as the mean from 3 technical replicates.

The solubility of oxygen was assumed to be 223 μmol O2·l^−1^ at 30 °C according to the Strathkelvin instruments manual. Respiration rates are expressed as nmol/min/mg protein.

Mitochondria were checked for leakage by the addition of 2.5 µM cytochrome c and 0.05 µg/mL oligomycin. The absence of an increase in flux after the addition of cytochrome c indicated the integrity of the mitochondrial outer membrane. When ATP synthesis was inhibited by oligomycin, the mitochondria were transferred to the state 2, which reflects the respiration rate compensating for the proton leak. These results indicate that the inner membrane is intact, and mitochondria were not damaged through the preparation procedure.

### 4.5. Statistical Analysis

The a priori power analysis (G*Power Version 3.1.7, University of Duesseldorf, Germany) with n = 10 animals per group at a given α ≤ 0.05 (two-tailed) and an expected difference in µHbO_2_ of at least 20% with an expected standard deviation from 10% to 15% (based on previous studies) suggested a power of 84.5%.

Normal distribution of microcirculatory and mitochondrial variables was confirmed in Q–Q plots or the Kolmogorov–Smirnov test, respectively. Microcirculatory data were analyzed with a two-way ANOVA for repeated measures, followed by Tukey’s post hoc test for differences between groups, and Dunnett’s post hoc test for differences versus baseline. Mitochondrial data were analyzed with the Kruskal–Wallis test with Dunn’s post hoc correction for multiple comparisons and presented as min/median/max (all statistical tests via GraphPad software v 10.1., Int, La Jolla, CA, USA).

Microcirculatory data are presented as means ± SD; mitochondrial data were presented as a box plot (min/median/max), and *p* < 0.05 was considered significant. Wherever delta values are presented, the absolute baseline value was subtracted from the absolute value at the respective observation points (30, 60 min) to individualize the data to each animal’s baseline. Therefore, the microcirculatory oxygenation data [ΔµHbO_2_] are provided as absolute percentage points with regard to baseline values, and the microcirculatory perfusion data [ΔµFlow] are provided as arbitrary units (AU).

## 5. Conclusions

This exploratory, randomized, placebo-controlled, blinded animal trial showed that modifying the autonomic balance in sepsis deteriorates intestinal microcirculation, especially in combination with NG. By contrast, the activation of the parasympathetic nervous system after sterile laparotomy can attenuate a vasodilator-induced decrease in microcirculatory oxygenation. Interestingly, despite the negative effect of the topically applied vasodilators on the microcirculation, Ilo showed rather positive effects on colonic mitochondrial function under non-septic conditions. Furthermore, we could demonstrate that a local therapy with vasodilating and parasympathetic acting agents has no effect on the macrohemodynamic variables.

Further studies are needed to reveal the underlying mechanisms, as well as clinical studies to investigate the effect of vasoactive drugs on intestinal microcirculation under septic and physiological conditions. Moreover, studies regarding different types of administration and their influence on the microcirculation in the splanchnic area could be of major interest.

## Figures and Tables

**Figure 1 ijms-25-09305-f001:**
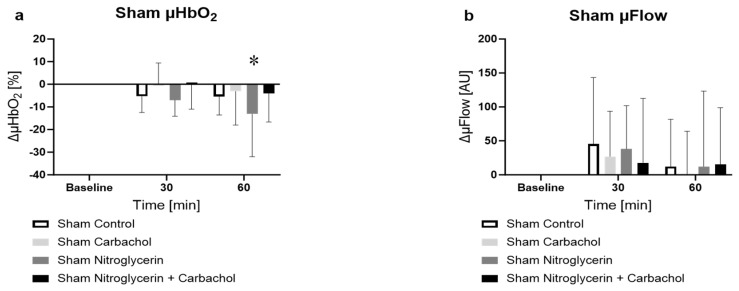
Effect of vehicle, carbachol, nitroglycerin (NG) or carbachol + NG on colonic microcirculatory (**a**) oxygenation [%] and (**b**) flow [AU] in non-septic animals. * = *p* < 0.05 versus baseline (two-way ANOVA with Dunnett post hoc correction, calculated for all sham groups, NG and Ilo shown separately for better distinctness), n = 10.

**Figure 2 ijms-25-09305-f002:**
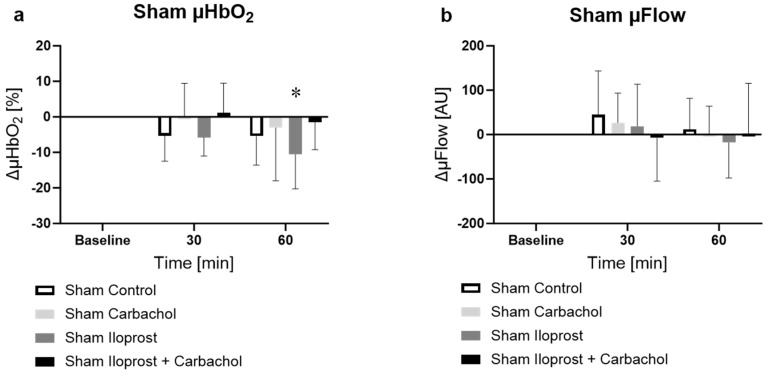
Effect of vehicle, carbachol, iloprost (Ilo) or carbachol + Ilo on colonic microcirculatory (**a**) oxygenation [%] and (**b**) flow [AU] in non-septic animals. * = *p* < 0.05 versus baseline (two-way ANOVA with Dunnett post hoc correction calculated for all sham groups, NG and Ilo shown separately for better distinctness), n = 10.

**Figure 3 ijms-25-09305-f003:**
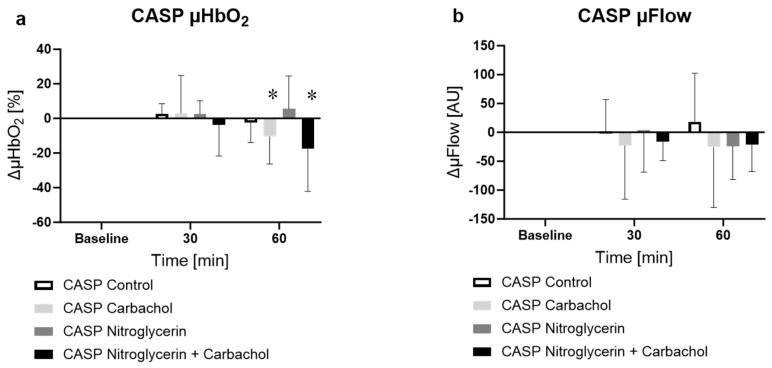
Effect of vehicle, carbachol, NG or carbachol + NG on colonic microcirculatory (**a**) oxygenation [%] and (**b**) flow [AU] in septic animals. * = *p* < 0.05 versus baseline (two-way ANOVA with Dunnett post hoc correction, calculated for all CASP groups, NG and Ilo shown separately for better distinctness), n = 10.

**Figure 4 ijms-25-09305-f004:**
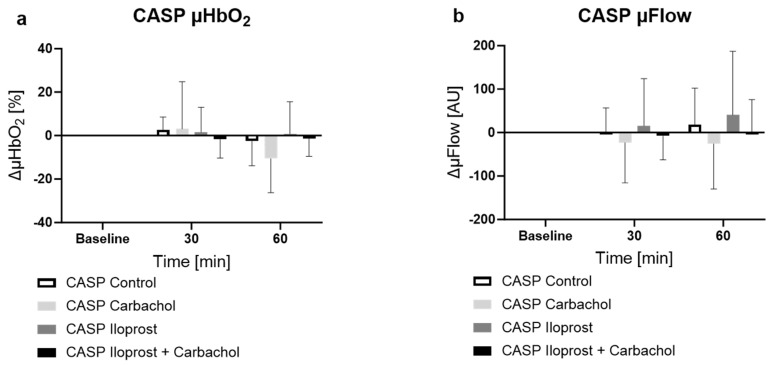
Effect of vehicle, carbachol, iloprost (Ilo) or carbachol + Ilo on colonic microcirculatory (**a**) oxygenation [%] and (**b**) flow [AU] in septic animals. * = *p* < 0.05 versus baseline (two-way ANOVA with Dunnett post hoc correction calculated for all CASP groups, NG and Ilo shown separately for better distinctness), n = 10.

**Figure 5 ijms-25-09305-f005:**
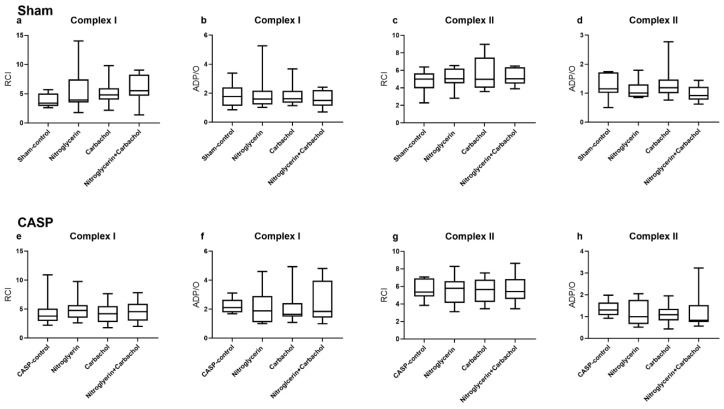
Effect of nitroglycerin, carbachol and combination of both drugs on mitochondrial respiration expressed as RCI (**a**,**c**,**e**,**g**) and ADP/O (**b**,**d**,**f**,**h**) after stimulation of the oxidative chain through complex I (**a**,**b**,**e**,**f**) and II (**c**,**d**,**g**,**h**) in colon under septic (**e**–**h**) and non-septic (**a**–**d**) conditions. Data are presented as min/median/max, n = 10, Kruskal–Wallis test with Dunn’s post hoc correction for multiple comparisons calculated for all CASP and sham groups, nitroglycerin and iloprost shown separately for better distinctness.

**Figure 6 ijms-25-09305-f006:**
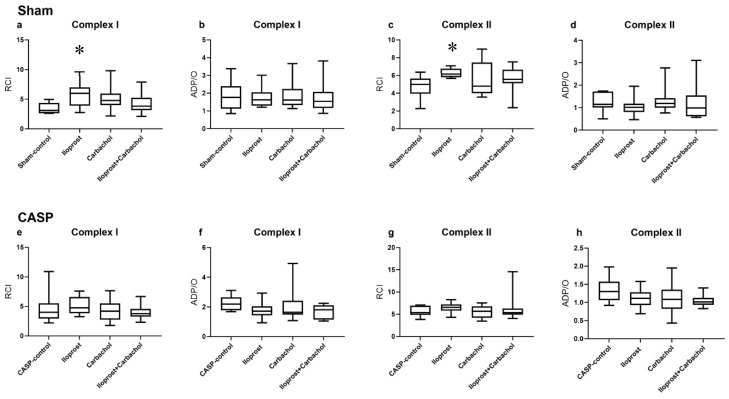
Effect of iloprost, carbachol and combination of both drugs on mitochondrial respiration expressed as RCI (**a**,**c**,**e**,**g**) and ADP/O (**b**,**d**,**f**,**h**) after stimulation of the oxidative chain through complex I (**a**,**b**,**e**,**f**) and II (**c**,**d**,**g**,**h**) in colon under septic (**e**–**h**) and non-septic (**a**–**d**) conditions. Data are presented as min/median/max, n = 10, Kruskal–Wallis test with Dunn’s post hoc correction for multiple comparisons calculated for all CASP and sham groups, nitroglycerin and iloprost shown separately for better distinctness.

**Figure 7 ijms-25-09305-f007:**
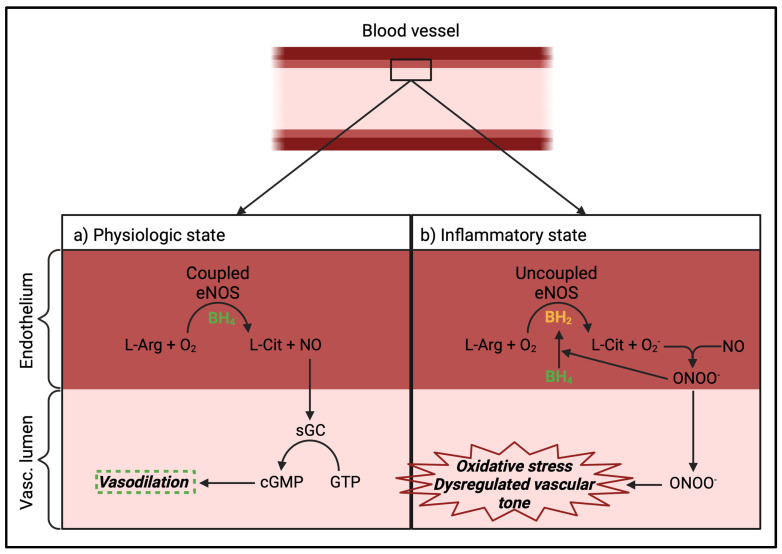
Graphic representation of eNOS uncoupling. (**a**) Physiologic state: in presence of cofactor BH_4_, NO is synthesized from L-Arg and oxygen by eNOS and leads to vascular dilation by activating the soluble guanylate cyclase. (**b**) Inflammatory state: in an inflammatory surrounding, oxidative stress causes absence of BH_4_ by oxidizing it to BH_2_, leading to the production of ONOO^−^. Abbreviations: eNOS: endothelial Nitrogenoxide-synthase; BH_4_: Tetrahydrobiopterin; BH_2_: Dihydrobiopterin; L-Arg: L-Arginine; L-Cit: L-Citrulline; NO: nitric oxide; O_2_: oxygen; O_2_^−^: superoxide; ONOO^−^: peroxynitrite; sGC: soluble guanylate cyclase; cGMP: cyclic guanosine monophosphate; GTP: guanosine triphosphate; Vasc.: vascular. Created with BioRender.com.

**Figure 8 ijms-25-09305-f008:**
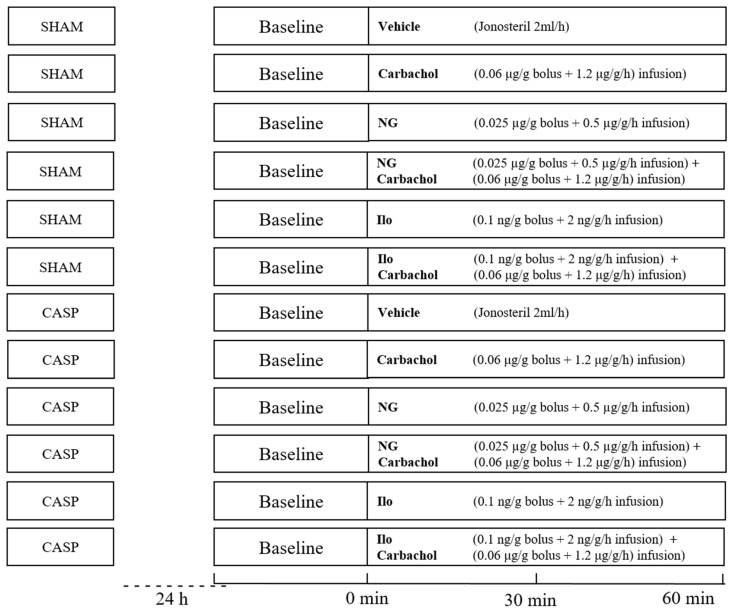
Experimental protocol. CASP and sham surgery were performed 24 h prior to catheterization and administration of topical therapy. CASP: colon ascendens stent peritonitis; NG: nitroglycerin; Ilo: Iloprost.

**Table 1 ijms-25-09305-t001:** Effect of topical therapy on hemodynamic parameters in sham and CASP groups, data are mean ± SD * *p* < 0.05 vs. baseline (two-way ANOVA with Dunnett post hoc correction) (n = 10 per group), HR = heart rate; MAP = mean arterial pressure; CASP: colon ascendens stent peritonitis; NG = nitroglycerine; Ilo = iloprost.

**Sham**	**Control**	**Carbachol**	**NG**	**NG + Carbachol**	**Ilo**	**Ilo + Carbachol**
HR [1·min^−1^]						
Baseline	490 ± 40	508 ± 52	472 ± 61	483 ± 60	490 ± 67	525 ± 35
30	459 ± 48	488 ± 38	456 ± 61	477 ± 85	483 ± 73	514 ± 49
60	435 ± 59 *	450 ± 48 *	429 ± 61 *	428 ± 73 *	463 ± 72	484 ± 45 *
MAP [mmHg]						
Baseline	121 ± 19	115 ± 25	121 ± 31	112 ± 28	141 ± 14	139 ± 29
30	105 ± 25	103 ± 27	102 ± 31	96 ± 42	129 ± 27	115 ± 35 *
60	91 ± 33 *	87 ± 20 *	100 ± 34 *	81 ± 40 *	109 ± 30 *	106 ± 37 *
**CASP**	**Control**	**Carbachol**	**NG**	**NG + Carbachol**	**Ilo**	**Ilo + Carbachol**
HR [1·min^−1^]						
Baseline	543 ± 48	484 ± 35	526 ± 61	507 ± 42	535 ± 55	526 ± 51
30	510 ± 65 *	466 ± 45 *	480 ± 53 *	496 ± 52 *	518 ± 58	516 ± 57
60	477 ± 62 *	456 ± 61 *	477 ± 48 *	485 ± 69 *	481 ± 57 *	500 ± 63
MAP [mmHg]						
Baseline	117 ± 33	102 ± 25	111 ± 32	99 ± 28	125 ± 26	125 ± 30
30	101 ± 34 *	97 ± 26	98 ± 34 *	88 ± 28	106 ± 26 *	112 ± 31
60	101 ± 34 *	97 ± 23	97 ± 43 *	77 ± 20 *	99 ± 27 *	106 ± 32 *

## Data Availability

The data presented in this study are available upon request from the corresponding author.

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
