# Peer review of "Effects of Local Vasodilators and the Autonomic Nervous System on Microcirculation and Mitochondrial Function in Septic Rats"

_ijms, 2024, doi:10.3390/ijms25179305_

Round 1

Reviewer 1 Report (New Reviewer)

Comments and Suggestions for Authors

This study examines the impact of topical vasodilators (nitroglycerin and iloprost) and the parasympathetic drug carbachol on colonic microcirculation and mitochondrial respiration in a sepsis model. In sham animals, nitroglycerin and iloprost reduced microcirculatory oxygenation, but carbachol mitigated this effect. In septic conditions, carbachol alone or combined with nitroglycerin decreased oxygenation, with the effects varying based on the pathophysiological state.

Overall, the manuscript is well-written and comprehensive. I have no significant criticisms, only a few minor suggestions for the authors to consider:

1. Fig 1-4: The Error bars are too high in all the figures even with total samples 10. What test was used; parametric or non-parametric? Why not One-way ANOVA for statistical significance?

2. Why were different tests for significance used for Fig1-4 and Fig5-6? There should be consistency between the significance tests.

Author Response

Comment 1. Fig 1-4: The Error bars are too high in all the figures even with total samples 10. What test was used; parametric or non-parametric? Why not One-way ANOVA for statistical significance?

Response 1: In the present study, for microcirculatory data we used a parametric 2-way ANOVA to examine the effect of our interventions at different time points between the groups and in comparison to the baseline. We were able to verify the normal distribution of our results via QQ plots, however, as is often the case in experimental animal studies, our results had a certain degree of scatter. Therefore, our measurements showed a difference between the individual groups and the baseline (which we calculated with the Dunnett post-hoc analysis), but no significant difference between the groups (which we calculated with the Tukey post-hoc analysis). Since our primary experimental question was how the substances affect the microcirculation and not which substance has a stronger or weaker influence in direct comparison, we focused primarily on this. For the data describing mitochondrial function we used Kruskal Wallis-Test with Dunnet’s post hoc correction, because non-parametric distribution was showed by Kolmogorow-Smirnow-Test.

Comment 2. Why were different tests for significance used for Fig1-4 and Fig5-6? There should be consistency between the significance tests.

Response 2: Thank you for the comment. Due to the fact that, in contrast to the microcirculation measurements, we only had one measurement point for the mitochondrial measurements and not three, we decided to use the Kruskal-Wallis' test. However, we consider both tests to be appropriate for evaluating the respective data set.

Reviewer 2 Report (New Reviewer)

Comments and Suggestions for Authors

The present paper of Marcus C et.al. investigated the effects of local vasodilators and the autonomic nervous system on microcirculation and mitochondrial function in an animal model of sepsis. Authors underwent an exploratory study aimed to investigate the effects of topical administered vasodilators and the parasympathetic drug carbachol on colon microcirculatory parameters.

In this aspect, the present paper is novel and interesting, thus worth to get published, Results are important and interesting because microcirculation disturbances are major pathophysiological aspects of sepsis leading to the failure of traditional treatments. Authors highlight into the pathophysiology of microcirculation, giving important conclusions on the effect of vasodilators and their action’s modification by parasympathetic nervous system.

Authors have done a good job. The article is well organized into different sections, methods are very well applied to answer the questions and data are clearly presented. Tables and figures are all with proper content supporting results with a clear manner. References are more than enough and appropriate to support the study. The discussion is unbiased. The paper is novel as other studies on NG and Ilo investigating the effect on mitochondrial function in colon of septic animals are lacking

I propose the acceptance of the current manuscript.

Author Response

Reviewer 2: The present paper of Marcus C et.al. investigated the effects of local vasodilators and the autonomic nervous system on microcirculation and mitochondrial function in an animal model of sepsis. Authors underwent an exploratory study aimed to investigate the effects of topical administered vasodilators and the parasympathetic drug carbachol on colon microcirculatory parameters.

In this aspect, the present paper is novel and interesting, thus worth to get published, Results are important and interesting because microcirculation disturbances are major pathophysiological aspects of sepsis leading to the failure of traditional treatments. Authors highlight into the pathophysiology of microcirculation, giving important conclusions on the effect of vasodilators and their action’s modification by parasympathetic nervous system.

Authors have done a good job. The article is well organized into different sections, methods are very well applied to answer the questions and data are clearly presented. Tables and figures are all with proper content supporting results with a clear manner. References are more than enough and appropriate to support the study. The discussion is unbiased. The paper is novel as other studies on NG and Ilo investigating the effect on mitochondrial function in colon of septic animals are lacking

I propose the acceptance of the current manuscript.

Response: We highly appreciate the feedback on our work.

Reviewer 3 Report (New Reviewer)

Comments and Suggestions for Authors

In this manuscript, author investigated the effects of topical administered vasodilators and the parasympathetic drug carbachol on colonic microcirculatory oxygenation (µHbO2), blood flow (µflow) and mitochondrial respiration. Most of results were interesting to understand the influence of topical application of direct vasodilative agents and the effect of a modification of the autonomic balance by a local acting parasympathomimetic on the gastrointestinal oxygenation as well as colonic mitochondrial function under septic conditions. The data are comprehensive and well presented in the figures, and the experimental approaches appear sound. However, the manuscript needs minor modifications to be considered by International Journal of Molecular Sciences.

Minor comments:

1)      First of all, text should be reviewed and corrected by native speaker.

2)      Abstracts should be clearly represented Background, Purpose, Methods, Results, and Conclusions.  

3)      Also, all abbreviation should be fully described when it firstly appeared. Also, this description should be not repeated in text. Ex) Table 1, Tab. 1/ Fig 1, Fig. 1

4)      All number should be separated unit except % and oC. Also, unit should be described same pattern.

5)      The name of each group was maintained the same form in all text, figure and figure legends. The name of the group is too long and complicated for readers to understand

6)      Figure legends should be corrected to include title, information for figure, and information for abbreviation.

7)      Author should describe the detail information about animal experiment including origin of animals, the number of animals per each group, diet information, breeding condition, anesthesia, euthanasia observation period et al.

8)      All reagents and instruments should be sequentially described Company, State (Country), Nation.

9)      Figure 8 was very complex to understand reader. Author should be considered correction of it.

10)   All number should be separated unit except % and oC. Also, unit should be described same pattern.

11)  References should be corrected according to journal guideline.

12)  Why did you select their doses? Author should describe a reason into Results and Discussion.

Comments on the Quality of English Language

Text should be reviewed and corrected by native speaker.

Author Response

Comment 1: First of all, text should be reviewed and corrected by native speaker.

Response 1: Thank you for the remark, the text has been revised in grammar and language.

Comment 2: Abstracts should be clearly represented Background, Purpose, Methods, Results, and Conclusions. 

Response 2: According to the IJMS Authors Instructions, “The abstract should be a total of about 200 words maximum. The abstract should be a single paragraph and should follow the style of structured abstracts, but without headings“(IJMS | Instructions for Authors (mdpi.com)) we did not include headings in the abstract to subdivide the abstract more precisely. In that form, the structure of the abstract sufficiently fulfills the formal requirements

Comment 3: Also, all abbreviation should be fully described when it firstly appeared. Also, this description should be not repeated in text. Ex) Table 1, Tab. 1/ Fig 1, Fig. 1

Response 3: We corrected the description in table 1 and Fig 1, furthermore we changed the repetitions in text (NG and Ilo instead of nitroglycerine and Iloprost) (See ll. xxx

Comment 4: All number should be separated unit except % and oC. Also, unit should be described same pattern.

Response 4: Numbers were corrected.

Comment 5: The name of each group was maintained the same form in all text, figure and figure legends. The name of the group is too long and complicated for readers to understand

Response 5: We shortened the group names in Table 1 and continued these designations in the course of the text. (See ll.

Comment 6: Figure legends should be corrected to include title, information for figure, and information for abbreviation.

Response 6: Figure legends were revised, all abbreviations are included. (See

Comment 7: Author should describe the detail information about animal experiment including origin of animals, the number of animals per each group, diet information, breeding condition, anesthesia, euthanasia observation period et al.

Response 7: We added information regarding the animal euthanasation process. (See ll 418 ff.). The rest of the information about breeding condition, numbers per group etc is available in part 4.1 (ll 381 ff. and ll 411 ff.)

Comment 8: All reagents and instruments should be sequentially described Company, State (Country), Nation.

Response 8: The company names and their Headoffice location were added (See ll 413, 422, 437, 440)

Comment 9: Figure 8 was very complex to understand reader. Author should be considered correction of it.

Response 9: The illustration has been revised, the abbreviations have been harmonised with the entire text and the font size of the dosages has been increased. (See ll 393 ff.)

Comment 10: All number should be separated unit except % and oC. Also, unit should be described same pattern.

Response 10: Compare Response 4.

Comment 11:  References should be corrected according to journal guideline.

Response 11: The references were corrected according to the IJMS manuscript standard (IJMS | Instructions for Authors (mdpi.com)

Comment 12: Why did you select their doses? Author should describe a reason into Results and Discussion.

Response 12: The doses of all drugs based on data from the literature. In the case of NG and Ilo we used adapted doses from trials carried out on dogs under hemorrhagic schock  (Truse et al. 2017, PMID: 28441653). In the case of carbachol, the dosis based on the study of Bao et al. who showed that carbachol improves microcirculatory flow in shock after burns in rats (Bao et al. 2010, PMID: 20061857). We added this information to the both chapters (See ll 371-376 and 469-470)

Reviewer 4 Report (New Reviewer)

Comments and Suggestions for Authors

Dear the Editor
Marcus C et al examined the physiological effect of NO generator on microcirculation in rat model. Colon ascendens stent peritonitis (CASP) model is an established model which allows measurement of colonic oxygenation, blood flow, and mitochondrial respiration. These authors uncovered that carbachol, a cholinergic agonist, reduced uHbO2 in microcirculation both alone and with nitroglycerin, indicating that these agents act as vasodilator in this experimental settings. In method section, surgical procedure was well-described. Although an initial examination of vasodilation was mainly performed using porcine aorta in three decades ago (so called a ring study), an experimental example using small animals was rather technically difficult. In this sense, these authors nicely demonstrate the role of NO-related agents in this rat model.

Major concerns:
None. Fine experiment.

Minor concerns:
In L473, some characters need to be subscribed (ie KH2PO4 etc).

Author Response

Major concerns:

None. Fine experiment.

Minor concerns:

Comment 1: In L473, some characters need to be subscribed (ie KH2PO4 etc).

Response 1: The proposed corrections were made (line 473+486), we highly appreciate your comments on our work.

This manuscript is a resubmission of an earlier submission. The following is a list of the peer review reports and author responses from that submission.

Round 1

Reviewer 1 Report

Comments and Suggestions for Authors

The authors investigated the effects of topically applied vasodilators and the parasympathetic drug carbachol on colonic microcirculatory oxygenation (µHbO2), blood flow (µflow), and mitochondrial respiration in 120 male Wistar rats with colonic ascending stent peritonitis (CASP) or sham surgery. The manuscript present extremely inovative topic, structured in standart English and written well. 

Minor coments:

1. The title- need to be shorten; to be add a graphical abstract

2. In introduction - With a prevalence of...............to be change

3. row 62-64 to be rewritten

4. parts 4.1 Ethic approval/ 4.2 Surgical induction of sepsis -  combined in one paragraph

5. row 384-388  ..... row 411- 420.....to be shorten; 

6. 4.4. Preparation of colon/liver homogenates and protein levels measurement

7. Results are well presented, without unnecessary additions. Table and figures presented well, in normal resolution, and the description is comprehensive.

8. The discussion part combined important results. Presenting a graphical mechanism of action of NG on endothelial dysfunction in a septic environment and uncoupling endothelial eNOS to its physiological mechanism, the production of superoxide anions (O2-) instead of NO............. is mandatory, will strengthen the findings of the study.

Тo suggest a graphical version of a probable mechanism of action by which iloprost improves the coupling between the respiratory chain and oxidative phosphorylation requires further justification.

9. to add study limitations and future work directions.

10. the conclusion part is written well.

11. The references from the last 4 years are only 21%, where it is possible to cite new studies.

Comments on the Quality of English Language

Minor editing of English language required

Author Response

Comment 1: The title- need to be shorten; to be add a graphical abstract

Response 1: We shortened the title while keeping the key information. A graphical abstract was added and uploaded separately.

Comment 2: In introduction - With a prevalence of...............to be change

Response 2: We have rewritten the sentence (page 1 line 26-28)

Comment 3: row 62-64 to be rewritten

Response 3: We changed the paragraph to eliminate the redundancy (page 1, line 61-63)

Comment 4: parts 4.1 Ethic approval/ 4.2 Surgical induction of sepsis -  combined in one paragraph

Response 4: We combined both subsections into one paragraph as requested (line 383-388).

Comment 5: row 384-388  ..... row 411- 420.....to be shorten; 

Response 5: Both parts were shortened  as suggested (page 12, line 407-411 and page 13, line 433-440).

Comment 6: 4.4. Preparation of colon/liver homogenates and protein levels measurement

Response 6: We changed the subheader and added protein levels to make the section more precise and easier to understand for the reader (line 472)

Comment 7: Results are well presented, without unnecessary additions. Table and figures presented well, in normal resolution, and the description is comprehensive.

Response 7: We appreciate the comment.

Comment 8: The discussion part combined important results. Presenting a graphical mechanism of action of NG on endothelial dysfunction in a septic environment and uncoupling endothelial eNOS to its physiological mechanism, the production of superoxide anions (O2-) instead of NO............. is mandatory, will strengthen the findings of the study. Тo suggest a graphical version of a probable mechanism of action by which iloprost improves the coupling between the respiratory chain and oxidative phosphorylation requires further justification.

Response 8: A graphical illustration of the mechanism of eNOS uncoupling was added. (page 9, line 293-301). The coupling between the respiratory chain and the oxidative phosphorylation was improved because the state 2 was lowered (the change was not significant). State 3 for complex I was lowered (not significant either) and for complex II was similar to the control. RCI is a calculated value and the not significant changes in state 2 and state 3 led to a significant change in RCI. We added the explanation to the text (page 10, line 342-346) but we would rather avoid the graphical presentation.

Comment 9: to add study limitations and future work directions.

Response 9: Thank you for your valuable remark. We created a separated chapter “Future work directions and limitations”. We transferred the part of the discussion referring to the limitations of the study and complement it with the future work direction (page 11, lines 365-380)

Comment 10: the conclusion part is written well.

Response 10: We appreciate the comment.

Comment 11: The references from the last 4 years are only 21%, where it is possible to cite new studies.

Response 11: Thank you for this remark. Citations were updated where possible. There is only few literature regarding nitroglycerine and iloprost and intestinal barrier function during sepsis, so in some points we let the original sources instead of citing the newer publications referring to the original manuscripts.

Reviewer 2 Report

Comments and Suggestions for Authors

Dear Editor/Authors,

Manuscript ID: ijms-3056576 entitled: "The effects of local acting vasodilators and modulation of the autonomic nervous system on colonic microcirculation and mi
tochondrial function in septic rats
" by authors: Marcus C., Hansen C., Schulz J., Hof S., Kuebart A., Truse R., Vollmer C., Bauer I., Picker O. and Herminghaus A present a study on the the possible effects of topical application of direct vasodilators and the effect of altering autonomic balance by a locally acting parasympathomimetic on gastrointestinal oxygenation and colonic mitochondrial function under septic conditions. Positive effects of topical therapy in the gastrointestinal tract could represent a therapeutic option, e.g. as an adjunct to enteral nutrition or as oral administration in the treatment of septic patients in the intensive care unit. Authors concluded that a local therapy with vasodilating and parasympathetic acting agents has no effect on the macro hemodynamic variables. Authors also show ​that activation of the parasympathetic nervous system can attenuate a decrease in microcirculatory oxygenation induced by vasodilators.    

Title
The title is clear and understandable for the readers.

Abstract
Abstract is mainly appropriate, corrections are needed. See text, Lines 18, 19-22.

Introduction

The introduction is correctly written. The authors explain the basics of sepsis, the changes that occur at the level of the GI circulation. They hypothesized that topical application of parasympathomimetics may have beneficial effects on the precapillary vasoconstriction that occurs in the sepsis, in order to avoid the systemic effects of these preparations on the cardiovascular system.

Results

The quality and presentation of the results are good. The text part is also good.

Corrections:

Line 160: Introduce full name. First mention in the manuscript

Based on the SDs shown in Figs. 1-4, it can be seen that the results obtained are too heterogeneous. In this case, it is questionable how the authors came to the conclusion that the results have a normal distribution.

Tab. 1: There is no comparison between the groups. Many of the effects appear to be due to the action of charbachol.

Discussion

The discussion does not adequately explain the results obtained There is no explanation of the results obtained on the basis of molecular mechanisms. Data that the authors did not measure would be very helpful, e.g. parameters of energy metabolism (ATP, ADP, AMP, pyruvate, lactate, etc.).

From Line 192 to Line 209: should be moved to the end of the Results section as a summary of the results. The discussion should be limited to explaining the results obtained and comparing them with results from literature sources.

Line 211: Abbbreviation in parentheses (CASP) Lines: 246-248: Presumption. It is necessary to indicate whether there are similar data in the literature.

Most of the discussion consists of a repetition of the results obtained. There is no clear explanation for the results obtained and often the discussion is based on assumptions and speculation without a clear background in the literature.

Materials  and Methods

All animal experiments have been approved by the appropriate institutions. Appropriate statisticalal analysis for these type of analyses were performed.

All procedures were carried out to the highest standards and in an appropriate manner. Information on the mortality of the animals during the experiment is missing. The statistical analysis is appropriate.

Conclusions

The conclusions are too speculative. No clear conclusions can be drawn from the results obtained, as it appears that many of the effects are due to charbachol and not to NG and Ilo.

Decision

In my opinion, this manuscript does not meet the standards required for publication in the journal Int. J. Mol. Sci. The manuscript does not provide answers to the molecular mechanisms behind the results obtained. Furthermore, the literature used is quite old, as only about 25% of the references are from the last 5 years. This does not detract from the value of the results obtained, because any result that leads to the elucidation of the molecular mechanisms in sepsis represents a contribution to a better knowledge of this disease.

All other comments are given in the text.

General conclusion: Not acceptable for publication.

Author Response

Comment 1 (Abstract): Abstract is mainly appropriate, corrections are needed. See text, Lines 18, 19-22.

Response 1: We appreciate the comment, we changed the abstract and added a more clear labelling regarding significance level. (page 1, line 17-20)

Comment 2 (results): Line 160: Introduce full name. First mention in the manuscript

Response 2: The Full name of RCI and ADP/O were introduced. (page 6, line 158-159)

Comment 3: Based on the SDs shown in Figs. 1-4, it can be seen that the results obtained are too heterogeneous. In this case, it is questionable how the authors came to the conclusion that the results have a normal distribution.

Response 3: Thank you for the comment. Animal models, especially regarding a complex and diverse pathophysiological state like sepsis tend to show high heterogeneity. However, our results were analysed for a normal distribution using the Q-Q-plots and appropriate statistical tests were used (GraphPad software v 10.1., Int, La Jolla, CA).

Comment 4: Tab. 1: There is no comparison between the groups. Many of the effects appear to be due to the action of charbachol.

Response 4: The comparison between the groups was calculated, but not displayed on the graphs because of a lack of significance.

Comment 5 (Discussion): The discussion does not adequately explain the results obtained There is no explanation of the results obtained on the basis of molecular mechanisms. Data that the authors did not measure would be very helpful, e.g. parameters of energy metabolism (ATP, ADP, AMP, pyruvate, lactate, etc.).

Response 5: In our experiments, we used a high experimental effort to investigate the effect of different local therapies the microcirculatory parameters and mitochondrial function in abdominal sepsis. Since our experimental setting in combination with the drug interventions used is not described in the literature, the comparability to another results is highly limited. Our study design was focused on microcirculation and mitochondrial oxygen consumption. We therefore did not carry out any investigations into further energy metabolism. We are currently planning further experiments to investigate endothelial dysfunction and eNOS uncoupling in more detail, as described in the future work directions section (page 11, lines 365-380).

Comment 6 : From Line 192 to Line 209: should be moved to the end of the Results section as a summary of the results. The discussion should be limited to explaining the results obtained and comparing them with results from literature sources.

Response 6: Thank you for the constructive remark, we changed the position of this paragraph into the results section. (page 7, line 188-207)

Comment 7: Line 211: Abbbreviation in parentheses (CASP) 

Response 7: We added the abbreviations as requested. (page 7, line 209)

Comment 8: Lines: 246-248: Presumption. It is necessary to indicate whether there are similar data in the literature.

Response 8: We have substantiated our statements with evidence from the literature for the given passage. (page 8 line 252)

Comment 9: Most of the discussion consists of a repetition of the results obtained. There is no clear explanation for the results obtained and often the discussion is based on assumptions and speculation without a clear background in the literature.

Response 9: We thank for the valuable comment. We agree with the reviewer, that the interpretation of our results is speculative. As we didn’t analyse the metabolic status in detail, we cannot support our results refering to microcirculatory changes and mitochondrial oxygen consumptions with own obtained data. To strengthen our interpretation we refer to the citations from the literature even if the available literature on this field is limited.

Comment 10: (Materials an Methods): All animal experiments have been approved by the appropriate institutions. Appropriate statistical analysis for these type of analyses were performed.

Response 10: We thank you for this comment.

Comment 11: All procedures were carried out to the highest standards and in an appropriate manner. Information on the mortality of the animals during the experiment is missing. The statistical analysis is appropriate.

Response 11: Many thanks you for the constructive remark. We have added the mortality of the animals to the paper (page 11, line 390-392)

Comment 12 (Conclusion): The conclusions are too speculative. No clear conclusions can be drawn from the results obtained, as it appears that many of the effects are due to charbachol and not to NG and Ilo.

Response 12: Please refer to response 5 and 9. In our experiments, we could not detect any significant effect of carbachol application in the iloprost groups either in sepsis or in the sham groups overall. In contrast, in the nitroglycerine groups the effect of carbochol was present. Thus, the interpretation of the role of carbachol in our experiments is not quite unambiguous. See part of the discussion on page 10, lines 305-317.    

Comment 13 (Decision): In my opinion, this manuscript does not meet the standards required for publication in the journal Int. J. Mol. Sci. The manuscript does not provide answers to the molecular mechanisms behind the results obtained. Furthermore, the literature used is quite old, as only about 25% of the references are from the last 5 years. This does not detract from the value of the results obtained, because any result that leads to the elucidation of the molecular mechanisms in sepsis represents a contribution to a better knowledge of this disease.

Response 13: Thank you for your assessment. However, we hope that our work does fulfil the requirements of the special issue: "Sepsis: From Molecular Mechanisms, Pathophysiology to Novel Therapeutic Approaches 4.0" despite the lack of evidence of molecular mechanisms. In our study, we used an animal model to investigate new therapeutic approaches for a very complex pathological state like sepsis, with the focus on microcirculation and  mitochondrial function. This provided additional insight into the underlying pathomechanisms, particularly with regard to vasomotor function. However, this study certainly offers potential for follow-up studies to further enlighten the molecular pathways on this topic.

Comment 14: All other comments are given in the text.

Response 14: The comments in the text have been checked and included (compare response 1 and response 8)